# STATISTICAL INFERENCE FOR INDIVIDUAL FAIRNESS

**Subha Maity**
Department of Statistics
University of Michigan
smaity@umich.edu

**Songkai Xue**
Department of Statistics
University of Michigan
sxue@umich.edu

**Mikhail Yurochkin**
IBM Research
MIT-IBM Watson AI lab
mikhail.yurochkin@ibm.com

**Yuekai Sun**
Department of Statistics
University of Michigan
yuekai@umich.edu

## ABSTRACT

As we rely on machine learning (ML) models to make more consequential decisions, the issue of ML models perpetuating or even exacerbating undesirable historical biases (*e.g.* gender and racial biases) has come to the fore of the public's attention. In this paper, we focus on the problem of detecting violations of individual fairness in ML models. We formalize the problem as measuring the susceptibility of ML models against a form of adversarial attack and develop a suite of inference tools for the adversarial cost function. The tools allow auditors to assess the individual fairness of ML models in a statistically-principled way: form confidence intervals for the worst-case performance differential between similar individuals and test hypotheses of model fairness with (asymptotic) non-coverage/Type I error rate control. We demonstrate the utility of our tools in a real-world case study.

## 1 INTRODUCTION

The problem of bias in machine learning systems is at the forefront of contemporary ML research. Numerous media outlets have scrutinized machine learning systems deployed in practice for violations of basic societal equality principles (Angwin et al., 2016; Dastin, 2018; Vigdor, 2019). In response researchers developed many formal definitions of algorithmic fairness along with algorithms for enforcing these definitions in ML models (Dwork et al., 2011; Hardt et al., 2016; Berk et al., 2017; Kusner et al., 2018; Ritov et al., 2017; Yurochkin et al., 2020). Despite the flurry of ML fairness research, the basic question of assessing fairness of a given ML model in a *statistically principled* way remains largely unexplored.

In this paper we propose a statistically principled approach to assessing individual fairness (Dwork et al., 2011) of ML models. One of the main benefits of our approach is it allows the investigator to *calibrate* the method; *i.e.* it allows the investigator to prescribe a Type I error rate. Passing a test that has a guaranteed small Type I error rate is the usual standard of proof in scientific investigations because it guarantees the results are reproducible (to a certain degree). This is also highly desirable in detecting bias in ML models because it allows us to certify whether an ML model will behave fairly at test time. Our method for auditing ML models abides by this standard.

There are two main challenges associated with developing a hypothesis test for individual fairness. First, how to formalize the notion of individual fairness in an *interpretable* null hypothesis? Second, how to devise a test statistic and calibrate it so that auditors can control the Type I error rate? In this paper we propose a test motivated by the relation between individual fairness and adversarial robustness (Yurochkin et al., 2020). At a high-level, our approach consists of two parts:

1. generating unfair examples: by unfair example we mean an example that is similar to a training example, but treated differently by the ML models. Such examples are similar to adversarial examples (Goodfellow et al., 2014), except they are only allowed to differ from a training example in certain protected or sensitive ways.

2. summarizing the behavior of the ML model on unfair examples: We propose a loss-ratio based approach that is not only scale-free, but also interpretable. For classification problems, we propose a variation of our test based on the error rates ratio.

## 1.1 RELATED WORK

At a high level, our approach is to use the difference between the empirical risk and the distributionally robust risk as a test statistic. The distributionally robust risk is the maximum risk of the ML model on similar training examples. Here similarity is measured by a fair metric that encodes our intuition of which inputs should be treated similarly by the ML model. We note that DRO has been extensively studied in the recent literature (Duchi et al., 2016; Blanchet & Murthy, 2016; Hashimoto et al., 2018), however outside of the fairness context with the exception of Yurochkin et al. (2020); Xue et al. (2020). Yurochkin et al. (2020) focus on training fair or robust ML models instead of auditing ML models.

Xue et al. (2020) also use the difference between the empirical and distributionally robust risks as a test statistic, but their test is only applicable to ML problems with *finite* feature spaces. This limitation severely restricts the applicability of their test. On the other hand, our test is suitable for ML problems with *continuous* features spaces. We note that the technical exposition in Xue et al. (2020) is dependant on the finite feature space assumption and in this work we develop a novel perspective of the problem that allows us to handle continuous feature spaces.

## 2 GRADIENT FLOW FOR FINDING UNFAIR EXAMPLES

In this section, we describe a gradient flow-based approach to finding unfair examples that form the basis of our suite of inferential tools. Imagine an auditor assessing whether an ML model is fair or not. The auditor aims to detect violations of individual fairness in the ML model. Recall Dwork et al. (2011)'s definition of individual fairness. Let $\mathcal{X} \subset \mathbb{R}^d$ and $\mathcal{Y} \subset \mathbb{R}^d$ be the input and output spaces respectively, and $f : \mathcal{X} \to \mathcal{Y}$ be an ML model to audit. The ML model $f$ is known as individually fair if

$$d_y(f(x_1), f(x_2)) \leq L_{\text{fair}} d_x(x_1, x_2) \text{ for all } x_1, x_2 \in \mathcal{X} \tag{2.1}$$

for some Lipschitz constant $L_{\text{fair}} > 0$. Here $d_x$ and $d_y$ are metrics on $\mathcal{X}$ and $\mathcal{Y}$ respectively. Intuitively, individually fair ML model treats similar samples similarly, and the fair metric $d_x$ encodes our intuition of which samples should be treated similarly. We should point out that $d_x(x_1, x_2)$ being small does not imply $x_1$ and $x_2$ are similar in all aspects. Even if $d_x(x_1, x_2)$ is small, $x_1$ and $x_2$ may differ much in certain attributes, e.g., protected/sensitive attributes.

Before moving on, we comment on the choice of the fair metric $d_x$. This metric is picked by the auditor and reflects the auditor's intuition about what is fair and what is unfair for the ML task at hand. It can be provided by a subject expert (this is Dwork et al. (2011)'s original recommendation) or learned from data (this is a recent approach advocated by Ilvento (2019); Wang et al. (2019); Mukherjee et al. (2020)). Section 4 provides details of picking a fair metric in our empirical studies.

To motivate our approach, we recall the distributionally robust optimization (DRO) approach to training individually fair ML models (Yurochkin et al., 2020). Let $f : \mathcal{X} \to \mathcal{Y}$ be an ML model and $\ell(f(x), y) : \mathcal{Z} \to \mathbb{R}_+$ be any smooth loss (*e.g.* cross-entropy loss). To search for differential treatment in the ML model, Yurochkin et al. (2020) solve the optimization problem

$$\max_{P:W(P,P_n)\leq\epsilon} \int_{\mathcal{Z}} \ell(f(x), y) dP(z), \tag{2.2}$$

where $W$ is the Wasserstein distance on probability distributions on feature space induced by the fair metric, $P_n$ is the empirical distribution of the training data, and $\epsilon$ is a moving budget that ensures the adversarial examples are close to the (original) training examples in the fair metric. Formally, this search for differential treatment checks for violations of *distributionally robust fairness*.

**Definition 2.1** (distributionally robust fairness (DRF) (Yurochkin et al., 2020)). *An ML model* $h : \mathcal{X} \to \mathcal{Y}$ *is* $(\epsilon, \delta)$-*distributionally robustly fair (DRF) WRT the fair metric* $d_x$ *iff*

$$\sup_{P:W(P,P_n)\leq\epsilon} \int_{\mathcal{Z}} \ell(z, h) dP(z) - \int_{\mathcal{Z}} \ell(z, h) dP_n(z) \leq \delta. \tag{2.3}$$

The optimization problem (2.2) is an infinite-dimensional problem, but its dual is more tractable. Blanchet & Murthy show that the dual of (2.2) is

$$\max_{P:W(P,P_n)\leq\epsilon} \mathbb{E}_P[\ell(f(x),y)] = \min_{\lambda\geq 0}\{\lambda\epsilon + \mathbb{E}_{P_n}[\ell_\lambda^c(x,y)]\}, \tag{2.4}$$

$$\ell_\lambda^c(x_i,y_i) \triangleq \max_{x\in\mathcal{X}}\{\ell(f(x),y_i) - \lambda d_x^2(x,x_i)\}. \tag{2.5}$$

In practice, since (2.5) is highly non-convex in general, auditors use gradient-based optimization algorithm to solve it and terminate the algorithm after few iterations. As a result, one can not guarantee optimality of the solution. However, optimality is required to establish convergence guarantees for DRO algorithms. This issue is typically ignored in practice when developing training algorithms, e.g. as in Yurochkin et al. (2020), but it should be treated with care when considering limiting distribution of the related quantities required to calibrate a test. We note that Xue et al. (2020) needed discrete feature space assumption due to the aforementioned concern: when the feature space is discrete, it is possible to solve (2.5) optimally by simply comparing the objective value at all points of the sample space. In this paper we adapt theory to practice, i.e. analyze the limiting distribution of (2.5) optimized for a fixed number of gradient steps.

The effects of early termination can be characterized by a continuous-time approximation of adversarial dynamics, which we called *gradient flow attack*. Given a sample $(x_0,y_0)$, the gradient flow attack solves a continuous-time ordinary differential equation (ODE)

$$\begin{cases} \dot{X}(t) = \nabla_x\{\ell(f(X(t)),y_0) - \lambda d_x^2(X(t),x_0)\}, \\ X(0) = x_0, \end{cases} \tag{2.6}$$

over time $t \geq 0$. For fixed penalty parameter $\lambda$ and stopping time $T > 0$, $\Phi : \mathcal{X} \times \mathcal{Y} \to \mathcal{X}$ is the *unfair map*

$$\Phi(x_0,y_0) \triangleq X(T). \tag{2.7}$$

Here the map $\Phi$ is well-defined as long as $g(x) \triangleq \nabla_x\{\ell(f(x),y_0) - \lambda d_x^2(x,x_0)\}$ is Lipschitz, i.e., $\|g(x_1) - g(x_2)\|_2 \leq L\|x_1 - x_2\|_2$ for some $L > 0$. Under this assumption, the autonomous Cauchy problem (2.6) has unique solution and thus $\Phi : \mathcal{X} \times \mathcal{Y} \to \mathcal{X}$ is a one-to-one function. We call $\Phi$ an unfair map because it maps samples in the data to similar areas of the sample space that the ML model performs poorly on. We note that data in this case is an audit dataset chosen by the auditor to evaluate individual fairness of the given model. The audit data *does not* need to be picked carefully and could be simply an iid sample (e.g. testing data). The unfair map plays the key role as it allows us to identify areas of the sample space where model violates individual fairness, even if the audit samples themselves reveal no such violations.

In the rest of the paper, we define the test statistic in terms of the unfair map instead of the optimal point of (2.5). This has two main benefits:

1. **computational tractability:** evaluating the unfair map is computationally tractable because integrating initial value problems (IVP) is a well-developed area of scientific computing (Heath, 2018). Auditors may appeal to any globally stable method for solving IVP's to evaluate the unfair map.
2. **reproducibility:** the non-convex nature of (2.5) means the actual output of any attempts at solving (2.5) is highly depend on the algorithm and the initial iterate. By defining the test statistic algorithmically, we avoid ambiguity in the algorithm and initial iterate, thereby ensuring reproducibility.

Of course, the tractability and reproducibility of the resulting statistical tests comes at a cost: power. Because we are not exactly maximizing (2.5), the ability of the test statistic to detect violations of individual fairness is limited by the ability of (2.7) to find (unfair) adversarial examples.

## 2.1 EVALUATING THE TEST STATISTIC

Solving IVP's is a well-studied problem in scientific computing, and there are many methods for solving IVP's. For our purposes, it is possible to use any globally stable method to evaluate the unfair map. One simple method is the *forward Euler method* with sufficiently small step size. Let

$0 = t_0 < t_1 < \ldots < t_N = T$ be a partition of $[0, T]$, and denote the step size by $\eta_k = t_k - t_{k-1}$ for $k = 1, \ldots, N$. Initialized at $x^{(0)} = x_0$, the forward Euler method repeats the iterations

$$x^{(k)} = x^{(k-1)} + \eta_k \cdot \nabla_x \{\ell(f(x^{(k-1)}), y_0) - \lambda d_x^2(x^{(k-1)}, x_0)\} \tag{2.8}$$

for $k = 1, \ldots, N$. The validity of discretizations of the forward Euler method is guaranteed by the following uniform bounds.

**Theorem 2.2** (Global stability of forward Euler method)**.** *Consider the solution path* $\{X(t)\}_{0 \le t \le T}$ *given by* (2.6) *and the sequence* $\{x^{(k)}\}_{k=0}^N$ *given by* (2.8)*. Let the maximal step size be* $h = \max\{\eta_1, \ldots, \eta_N\}$*. Suppose that* $\|J_g(x)g(x)\|_\infty \le m$*, where* $g(x) = \nabla_x\{\ell(f(x), y_0) - \lambda d_x^2(x, x_0)\}$ *and* $J_g$ *is the Jacobian matrix of* $g$*. Then we have*

$$\max_{k=1,\ldots,N} \|X(t_k) - x^{(k)}\|_2 \le \frac{hm\sqrt{d}}{2L}(e^{LT} - 1). \tag{2.9}$$

Theorem 2.2 indicates that the global approximation error (2.9) decreases linearly with $h$, the maximal step size. Therefore by taking small enough $h$, the value of the unfair map $\Phi$ can be approximated by $x^{(N)}$ with arbitrarily small error.

## 3 TESTING INDIVIDUAL FAIRNESS OF AN ML MODEL

Although gradient flows are good ways of finding unfair examples, they do not provide an interpretable summary of the ML model outputs. In this section, we propose a loss-ratio based approach to measuring unfairness with unfair examples. Given a sample point $(x_0, y_0) \in \mathcal{Z}$, gradient flow attack (2.6) always increases the regularized loss in (2.5), that is,

$$\ell(f(x_0), y_0) \le \ell(f(X(T)), y_0) - d_x^2(X(T), x_0) \le \ell(f(X(T)), y_0). \tag{3.1}$$

Therefore the unfair map $\Phi : \mathcal{Z} \to \mathcal{X}$ always increases the loss value of the original sample. In other words, the ratio

$$\frac{\ell(f(\Phi(x, y)), y)}{\ell(f(x), y)} \ge 1 \text{ for all } (x, y) \in \mathcal{Z}. \tag{3.2}$$

Recall that the unfair map $\Phi$ moves a sample point to its similar points characterized by the fair metric $d_x$. The fair metric $d_x$ reflects the auditor's particular concern of individual fairness so that the original sample $(x, y)$ and the mapped sample $(\Phi(x, y), y)$ should be treated similarly. If there is no bias/unfairness in the ML model, then we expect the ratio $\ell(f(\Phi(x, y)), y)/\ell(f(x), y)$ to be close to 1. With this intuition, to test if the ML model is individually fair or not, the auditor considers hypothesis testing problem

$$H_0 : \mathbb{E}_P\left[\frac{\ell(f(\Phi(x, y)), y)}{\ell(f(x), y)}\right] \le \delta \quad \text{versus} \quad H_1 : \mathbb{E}_P\left[\frac{\ell(f(\Phi(x, y)), y)}{\ell(f(x), y)}\right] > \delta, \tag{3.3}$$

where $P$ is the true data generating process, and $\delta > 1$ is a constant specified by the auditor. Using the ratio of losses in (3.3) has two main benefits:

1. *scale-free*: changing the loss function by multiplying a factor does not change the interpretation of the null hypothesis.
2. The test is *interpretable*: the tolerance $\delta$ is the maximum loss differential above which we consider an ML model unfair. In applications where the loss can be interpreted as a measure of the negative impact of an ML model, there may be legal precedents on acceptable levels of differential impact that is tolerable. In our computational results, we set $\delta$ according to the four-fifths rule in US labor law. Please see Section 3.2 for further discussion regarding $\delta$ and interpretability.

We note that using the ratio of losses as the test statistic is not without its drawbacks. If the loss is small but non-zero, then the variance of the test statistic is inflated and and the test loses power, but Type I error rate control is maintained.

## 3.1 THE AUDIT VALUE

The auditor collects a set of audit data $\{(x_i, y_i)\}_{i=1}^n$ and computes the empirical mean and variance of the ratio $\ell(f(\Phi(x, y)), y) / \ell(f(x), y)$,

$$S_n = \frac{1}{n} \sum_{i=1}^n \frac{\ell(f(\Phi(x_i, y_i)), y_i)}{\ell(f(x_i), y_i)} \quad \text{and} \quad V_n^2 = \frac{1}{n-1} \sum_{i=1}^n \left( \frac{\ell(f(\Phi(x_i, y_i)), y_i)}{\ell(f(x_i), y_i)} - S_n \right)^2, \quad (3.4)$$

by solving gradient flow attack (2.6). The first two empirical moments, $S_n$ and $V_n^2$, are sufficient for the auditor to form confidence intervals and perform hypothesis testing for the population mean $\mathbb{E}_P[\ell(f(\Phi(x, y)), y) / \ell(f(x), y)]$, the *audit value*.

**Theorem 3.1** (Asymptotic distribution)**.** *Assume that* $\nabla_x \{\ell(f(x), y) - \lambda d_x^2(x, y)\}$ *is Lipschitz in* $x$, *and* $\ell(f(\Phi(x, y)), y) / \ell(f(x), y)$ *has finite first and second moments. If the ML model* $f$ *is independent of* $\{(x_i, y_i)\}_{i=1}^n$, *then*

$$\sqrt{n} V_n^{-1} \left( S_n - \mathbb{E}_P \left[ \frac{\ell(f(\Phi(x, y)), y)}{\ell(f(x), y)} \right] \right) \xrightarrow{d} \mathcal{N}(0, 1) \qquad (3.5)$$

*in distribution, as* $n \to \infty$.

The first inferential task is to provide confidence intervals for the audit value. The two-sided equal-tailed confidence interval for the audit value $\mathbb{E}_P[\ell(f(\Phi(x, y)), y) / \ell(f(x), y)]$ with asymptotic coverage probability $1 - \alpha$ is

$$\text{CI}_{\text{two-sided}} = \left[ S_n - \frac{z_{1-\alpha/2}}{\sqrt{n}} V_n, S_n + \frac{z_{1-\alpha/2}}{\sqrt{n}} V_n \right], \qquad (3.6)$$

where $z_q$ is the $q$-th quantile of normal distribution $\mathcal{N}(0, 1)$.

**Corollary 3.2** (Asymptotic coverage of two-sided confidence interval)**.** *Under the assumptions in Theorem 3.1,*

$$\lim_{n \to \infty} \mathbb{P} \left( \mathbb{E}_P \left[ \frac{\ell(f(\Phi(x, y)), y)}{\ell(f(x), y)} \right] \in \left[ S_n - \frac{z_{1-\alpha/2}}{\sqrt{n}} V_n, S_n + \frac{z_{1-\alpha/2}}{\sqrt{n}} V_n \right] \right) = 1 - \alpha. \qquad (3.7)$$

The second inferential task is to test restrictions on the audit value, that is, considering the hypothesis testing problem (3.3). Similar to the two-sided confidence interval (3.6), we may also have one-sided confidence interval for the audit value $\mathbb{E}_P[\ell(f(\Phi(x, y)), y) / \ell(f(x), y)]$ with asymptotic coverage probability $1 - \alpha$, i.e.,

$$\text{CI}_{\text{one-sided}} = \left[ S_n - \frac{z_{1-\alpha}}{\sqrt{n}} V_n, \infty \right). \qquad (3.8)$$

The one-sided confidence interval (3.8) allows us to test simple restrictions of the form

$$\mathbb{E}_P \left[ \frac{\ell(f(\Phi(x, y)), y)}{\ell(f(x), y)} \right] \leq \delta \qquad (3.9)$$

by checking whether $\delta$ falls in the $(1 - \alpha)$-level confidence region. By the duality between confidence intervals and hypothesis tests, this test has asymptotic Type I error rate at most $\alpha$. With this intuition, a valid test is:

$$\text{Reject } H_0 \text{ if } T_n > \delta, \quad T_n \triangleq S_n - \frac{z_{1-\alpha}}{\sqrt{n}} V_n, \qquad (3.10)$$

where $T_n$ is the test statistic.

**Corollary 3.3** (Asymptotic validity of test)**.** *Under the assumptions in Theorem 3.1,*

*1. if* $\mathbb{E}_P[\ell(f(\Phi(x, y)), y) / \ell(f(x), y)] \leq \delta$, *we have* $\lim_{n \to \infty} \mathbb{P}(T_n > \delta) \leq \alpha$;
*2. if* $\mathbb{E}_P[\ell(f(\Phi(x, y)), y) / \ell(f(x), y)] > \delta$, *we have* $\lim_{n \to \infty} \mathbb{P}(T_n > \delta) = 1$.

## 3.2 Test interpretability, test tolerance and an alternative formulation

To utilize our test, a auditor should set a tolerance (or threshold) parameter $\delta$. It is important that a auditor can understand and interpret the meaning of the threshold they choose. Appropriate choice of $\delta$ can vary based on the application, however here we consider a general choice motivated by the US Equal Employment Opportunity Commission's four-fifths rule, which states "selection rate for any race, sex, or ethnic group [must be at least] four-fifths (4/5) (or eighty percent) of the rate for the group with the highest rate".[1] Rephrasing this rule in the context of the loss ratio, we consider the following: the largest permissible loss increase on an individual should be at most five-fourth (5/4) of its original loss. This corresponds to the null hypothesis threshold $\delta = 1.25$.

The aforementioned wording of the four-fifth rule is based on the demographic parity group fairness definition, however it can be generalized to other group fairness definitions as follows: "performance of the model on any protected group must be at least four-fifth of the best performance across groups". Depending on what we mean by performance, we can obtain other group fairness definitions such as accuracy parity when auditing an ML classification system. In our test, we use the loss ratio because the loss is a general measure of performance of an ML system. However, in the context of supervised learning, the loss is often a mathematically convenient proxy for the ultimate quantity of interest, error rate. For classification problems, it is possible to modify our test statistic based on the ratio of error rates instead of losses (maintaining $\delta = 1.25$ according to the five-fourth rule).

Let $\ell_{0,1}$ be the 0-1 loss. Naively, we could consider the mean of the ratio $\frac{\ell_{0,1}(f(\Phi(x,y)),y)}{\ell_{0,1}(f(x),y)}$ as a test statistic, but this is problematic because the ratio is not well-defined when the classifier correctly classifies $x$. To avoid this issue, we propose considering the ratio of means (instead of the mean of the ratio) as a test statistic. Formally, we wish to test the hypothesis

$$H_0 : \frac{\mathbb{E}_P[\ell_{0,1}(f(\Phi(x,y)),y)]}{\mathbb{E}_P[\ell_{0,1}(f(x),y)]} \leq \delta \quad \text{versus} \quad H_1 : \frac{\mathbb{E}_P[\ell_{0,1}(f(\Phi(x,y)),y)]}{\mathbb{E}_P[\ell_{0,1}(f(x),y)]} > \delta, \qquad (3.11)$$

We emphasize that the gradient flow attack is still performed with respect to a smooth loss function; we merely use the 0-1 loss function to evaluate the accuracy of the classifier on the original and adversarial examples.

The auditor collects a set of audit data $\{(x_i, y_i)\}_{i=1}^n$ and computes the ratio of empirical risks

$$\tilde{S}_n = \frac{\frac{1}{n}\sum_{i=1}^n \ell_{0,1}(f(\Phi(x_i,y_i)),y_i)}{\frac{1}{n}\sum_{i=1}^n \ell_{0,1}(f(x_i),y_i)} \text{ and } \tilde{V}_n \triangleq \frac{1}{n}\sum_{i=1}^n \begin{bmatrix} \ell_{0,1}(f(\Phi(x_i,y_i)),y_i) \\ \ell_{0,1}(f(x_i),y_i) \end{bmatrix} \begin{bmatrix} \ell_{0,1}(f(\Phi(x_i,y_i)),y_i) \\ \ell_{0,1}(f(x_i),y_i) \end{bmatrix}^\top$$
(3.12)

by performing the gradient flow attack (2.6). Let $A_n$ and $B_n$ be the numerator and denominator of $\tilde{S}_n$. Parallel to the intuition of (3.10), here the proposed test is:

$$\text{Reject } H_0 \text{ if } \tilde{T}_n > \delta, \quad \tilde{T}_n \triangleq \tilde{S}_n - \frac{z_{1-\alpha}}{B_n^2}\sqrt{\frac{A_n^2(\tilde{V}_n)_{2,2} + B_n^2(\tilde{V}_n)_{1,1} - 2A_nB_n(\tilde{V}_n)_{1,2}}{n}}, \quad (3.13)$$

where $\tilde{T}_n$ is the test statistic. Please see Appendix C for the asymptotic normality theorem and Type I error guarantees.

## 4 Individual fairness testing in practice

In our experiments we first verify our methodology in simulations and then present a case-study of testing individual fairness on the `Adult` dataset (Dua & Graff, 2017).

A auditor performing the testing would need to make three important choices: fair metric $d_x(\cdot, \cdot)$, testing threshold $\delta$ to have a concrete null hypothesis and level of significance (maximum allowed type I error of hypothesis testing, i.e. $p$-value cutoff) to make a decision whether the null (classifier is fair) should be rejected. The fair metric can be provided by a subject expert, as considered in our simulation studies, or estimated from data using fair metric learning techniques proposed in the literature, as we do in the `Adult` experiment. Following the discussion in Section 3.2 we set

---

[1] Uniform Guidelines on Employment Selection Procedures, 29 C.F.R. §1607.4(D) (2015).

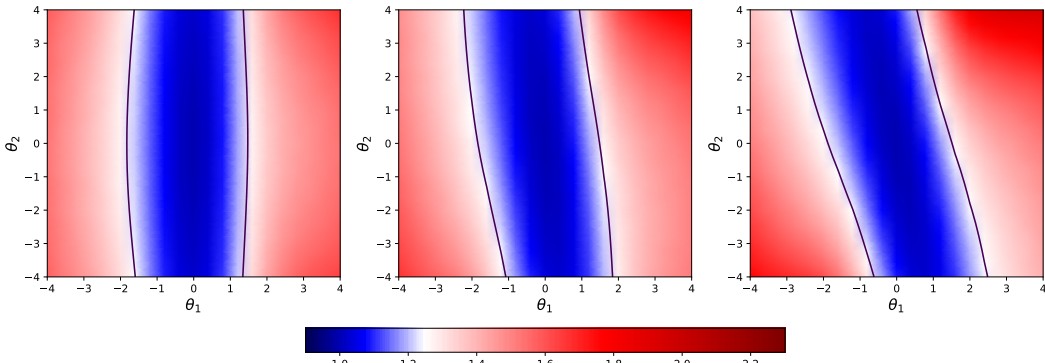

Figure 1: Heatmaps of the test statistic $T_n$ (3.10) for a logistic regression classifier on a grid of coefficients $(\theta_1, \theta_2)$. Individually fair classifier corresponds to $\theta_1$ close to 0 and any $\theta_2$. Black line in each plot represents the null hypothesis rejection decision boundary $T_n > 1.25$. Blue color represents acceptance region, whereas the red corresponds to unfair coefficients regions. The true fair metric discounts any differences along the first coordinate (left); (center,right) are results with misspecified fair metric, i.e. discounting direction is rotated by $5°$ and $10°$ respectively.

$\delta = 1.25$. For the significance level, typical choice in many sciences utilizing statistical inference is 0.05, which we follow in our experiments, however this is not a universal rule and should be adjusted in practice when needed.

## 4.1 STUDYING TEST PROPERTIES WITH SIMULATIONS

We first investigate the ability of our test to identify an unfair classifier, explore robustness to fair metric misspecification verifying Theorem A.1, and discuss implications of the choice of null hypothesis parameter $\delta$. We simulate a 2-dimensional binary classification dataset with two subgroups of observations that differ only in the first coordinate (we provide additional details and data visualization in Appendix E). One of the subgroups is underrepresented in the training data yielding a corresponding logistic regression classifier that overfits the majority subgroup and consequently differentiates data points (i.e. "individuals") along both coordinates. Recall that a pair of points that only differ in the first coordinate are considered similar by the problem design, i.e. their fair distance is 0, and prediction for such points should be the same to satisfy individual fairness.

**Fair metric** Our expert knowledge of the problem allow to specify a fair metric $d_x^2((x_1, x_2), (x_1', x_2')) = (x_2 - x_2')^2$. Evidently, an individually fair logistic regression should have coefficient of the first coordinate $\theta_1 = 0$, while intercept and $\theta_2$ can be arbitrary. The more $\theta_1$ differs from 0, the larger is the individual fairness violation. In Figure 1 (left) we visualize the heatmap of the test statistic (3.10) over a grid of $(\theta_1, \theta_2)$ values (intercept is estimated from the data for each coefficients pair). Recall that when this value exceeds $\delta = 1.25$ our test rejects the null (fairness) hypothesis (red heatmap area). Our test well-aligns with the intuitive interpretation of the problem, i.e. test statistic increases as $\theta_1$ deviates from 0 and is independent of the $\theta_2$ value.

**Metric misspecification** We also consider fair metric misspecification in the center and right heatmaps of Figure 1. Here the discounted movement direction in the metric is rotated, i.e. $d_x^2((x_1, x_2), (x_1', x_2')) = (\sin^2 \beta)(x_1 - x_1')^2 + (\cos^2 \beta)(x_2 - x_2')^2$ for $\beta = 5°$ (center) and $\beta = 10°$ (right). We see that the test statistic starts to reject fairness of the models with larger $\theta_2$ magnitudes due to misspecification of the metric, however it remains robust in terms of identifying $\theta_1 = 0$ as the fair model.

**Null hypothesis threshold** Finally we assess the null hypothesis choice $\delta = 1.25$. We saw that the test permits (approximately) $\theta_1 < 1.5$ — whether this causes minor or severe individual fairness violations depends on the problem at hand. A auditor that has access to an expert knowledge for defining the fair metric and desires stricter individual fairness guarantees may consider smaller values of $\delta$. In this simulated example, we see that as $\delta$ approaches 1, the test constructed with the correct fair metric (Figure 1 left) will reject all models with $\theta_1 \neq 0$, while permitting any $\theta_2$ values.

## 4.2 REAL DATA CASE-STUDY

We present a scenario of how our test can be utilized in practice. To this end, we consider income classification problem using `Adult` dataset (Dua & Graff, 2017). The goal is to predict if a person is earning over $50k per year using features such as education, hours worked per week, etc. (we exclude race and gender form the predictor variables; please see Appendix F and code in the supplementary materials for data processing details).

**Learning the fair metric** In lieu of an expert knowledge to define a fair metric, we utilize technique by Yurochkin et al. (2020) to learn a fair metric from data. They proposed a fair metric of the form: $d_x^2(x, x') = \langle x - x', P(x - x') \rangle$, where $P$ is the projection matrix orthogonal to a "sensitive" subspace. Similar to their `Adult` experiment, we learn this subspace by fitting two logistic regression classifier to predict gender and race and taking span of the coefficient vectors (i.e. vectors orthogonal to decision boundary) as the sensitive subspace. The intuition behind this metric is that this subspace captures variation in the data pertaining to the racial and gender differences. A fair metric should treat individuals that only differ in gender and/or race as similar, therefore it assigns 0 distance to any pair of individuals that only differ by a vector in the sensitive subspace (similar to the fair metric we used in simulations discounting any variation along the first coordinate). Our hypothesis test is an audit procedure performed at test time, so we learn the fair metric using test data to examine fairness of several methods that only have access to an independent train set to learn their decision function.

**Results** We perform testing of the 4 classifiers: baseline NN, group fairness Reductions (Agarwal et al., 2018) algorithm, individual fairness SenSR (Yurochkin et al., 2020) algorithm and a basic Project algorithm that pre-processes the data by projecting out "sensitive" subspace. For SenSR fair metric and Project we use training data to learn the "sensitive" subspace. All methods are trained to account for class imbalance in the data and we report test balanced accuracy as a performance measure following prior studies of this dataset (Yurochkin et al., 2020; Romanov et al., 2019). Results of the 10 experiment repetitions are summarized in Table 1 (see Table 3 in Appendix F.6 for the standard deviations). We compare group fairness using average odds difference (AOD) (Bellamy et al., 2018) for gender and race. Significance level for null hypothesis rejection is 0.05 and $\delta = 1.25$ (see Appendix F and code for details regarding the algorithms and comparison metrics).

Baseline exhibits clear violation of both individual ($T_n, \tilde{T}_n \gg 1.25$ and rejection proportion is 1) and group fairness (both AOD are far from 0). Simple projection pre-processing improved individual fairness, however the null is still rejected in the majority experiment repetitions (balanced accuracy improvement is accidental). A more sophisticated individual fairness algorithm SenSR does perform well according to our test with test statistic close to 1 (ideal value) and the test fails to reject individual fairness of SenSR every time. Lastly we examine the trade-off between individual and group fairness. Enforcing group fairness with Reductions leads to best AOD values, however it *worsens* individual fairness (comparing to the baseline) as measured by the test statistic. On the contrary, enforcing individual fairness with SenSR also improves group fairness metrics, however at the cost of the lowest balanced accuracy. We present a similar study of the COMPAS dataset in Appendix G. Results there follow the same pattern with the exception of Reductions slightly improving individual fairness in comparison to the baseline, but still being rejected by our test in all experiment repetitions.

Both loss-ratio test $T_n$ and error-rate ratio test $\tilde{T}_n$ results are almost identical. The only difference is that loss ratio test rejected Project in 9 out of 10 trials, while error-rate ratio test in 8 out of 10 trials.

**Setting stopping time $T$.** Recall that the test statistic $T_n$ depends on the number of steps $T$ of the gradient flow attack (2.6). Corollary 3.3 guarantees Type I error control for any $T$, i.e. it controls the error of rejecting a fair classifier regardless of the stopping time choice. Theoretical guarantees for Type II error, i.e. failing to reject an unfair classifier, are hard to provide in general (one needs to know expected value of the loss ratio for a given $T$ and a specific model). In practice, we recommend running the gradient flow attack long enough (based on the available computation budget) to guarantee small Type II error. In our `Adult` experiment we set $T = 500$. In Figure 2

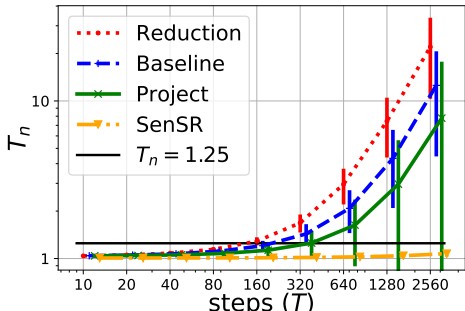

Figure 2: $T_n$ as a function of stopping time $T$ on `Adult` data (log-log scale)

Table 1: Results on `Adult` data over 10 experiment repetitions

|  | bal-acc | $AOD_{gen}$ | $AOD_{race}$ | Entropy loss | | 0-1 loss | |
|---|---|---|---|---|---|---|---|
|  |  |  |  | $T_n$ | rejection prop | $\tilde{T}_n$ | rejection prop |
| Baseline | 0.817 | -0.151 | -0.061 | 3.676 | 1.0 | 2.262 | 1.0 |
| Project | **0.825** | -0.147 | -0.053 | 1.660 | 0.9 | 1.800 | 0.8 |
| Reduction | 0.800 | **0.001** | **-0.027** | 5.712 | 1.0 | 3.275 | 1.0 |
| SenSR | 0.765 | -0.074 | -0.048 | **1.021** | **0.0** | **1.081** | **0.0** |

(note the log-log scale) we present an empirical study
of the test statistic $T_n$ as a function of stopping time $T$. We see that our test fails to reject SenSR, the classifier we found individually fair, for any value of $T$ verifying our Type I error guarantees in practice. Rejection of the unfair classifiers requires sufficiently large $T$, supporting our recommendation for Type II error control in practice.

## 5 DISCUSSION AND CONCLUSION

We developed a suite of inferential tools for detecting and measuring individual bias in ML models. The tools require access to the gradients/parameters of the ML model, so they're most suitable for internal investigators. We hope our tools can help auditors verify individual fairness of ML models and help researchers benchmark individual fairness algorithms. Future work on learning flexible individual fairness metrics from data will expand the applicability range of our test.

We demonstrated the utility of our tools by using them to reveal the gender and racial biases in an income prediction model. In our experiments, we discovered that enforcing group fairness may incur individual bias. In other words, the algorithm may sacrifice individual fairness in order to preserve parity of certain metrics across groups. For example, one of the earliest methods for enforcing group fairness explicitly treated examples from the majority and minority groups differently (Hardt et al., 2016). We conjecture that the even the more modern methods for enforcing group fairness could be forcibly balancing outcomes among demographic groups, leading to instances where similar individuals in different demographic groups are treated differently. The possible trade-off between individual and group fairness warrants further investigation, but is beyond the scope of this paper.

## ACKNOWLEDGEMENTS

This paper is based upon work supported by the National Science Foundation (NSF) under grants no. 1830247 and 1916271. Any opinions, findings, and conclusions or recommendations expressed in this paper are those of the authors and do not necessarily reflect the views of the NSF.

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

## A    ROBUSTNESS OF TEST STATISTIC TO THE CHOICE OF FAIR METRIC

Since the fair metric $d_x$ is picked by an expert's domain knowledge or in a data-driven way, the auditor may incur the issue of fair metric misspecification. Fortunately, the test statistic (3.10) is robust to small changes in fair metrics. Let $d_1, d_2 : \mathcal{X} \times \mathcal{X} \to \mathbb{R}_+$ be two different fair metrics on $\mathcal{X}$. Let $\Phi_1, \Phi_2 : \mathcal{X} \times \mathcal{Y} \to \mathcal{X}$ be the unfair maps induced by $d_1, d_2$. We start by stating the following assumptions:

(A1)  $\nabla_x\{\ell(f(x), y) - \lambda d_1^2(x, y)\}$ is $L$-Lipschitz in $x$ with respect to $\|\cdot\|_2$;
(A2)  $\sup_{x, x' \in \mathcal{X}} \|x - x'\|_2 \triangleq D < \infty$;
(A3)  $\ell(f(x), y)$ is $L_0$-Lipschitz in $x$ with respect to $\|\cdot\|_2$, and lower bounded by $c > 0$;
(A4)  $\sup_{x, x' \in \mathcal{X}} \|\nabla_x d_1^2(x, x') - \nabla_x d_2^2(x, x')\|_2 \leq D\delta_d$ for some constant $\delta_d \geq 0$.

Assumption A1 is always assumed for the existence and uniqueness of ODE solution. Assumption A2, that the feature space $\mathcal{X}$ is bounded, and the first part of Assumption A3 are standard in the DRO literature. The second part of Assumption A3 is to avoid singularity in computing loss ratio. Assumption A4 is worthy of being discussed. The constant $\delta_d$ in A4 characterizes the level of fair metric misspecification. Moreover, Assumption A4 is mild under Assumption A2. For example, let

$$d_1^2(x, x') = \langle x - x', \Sigma_{\text{exact}}(x - x')\rangle \text{ and } d_2^2(x, x') = \langle x - x', \Sigma_{\text{mis}}(x - x')\rangle \tag{A.1}$$

be the exact and misspecified fair metric respectively. Then

$$\sup_{x, x' \in \mathcal{X}} \|\nabla_x d_1^2(x, x') - \nabla_x d_2^2(x, x')\|_2 = \sup_{x, x' \in \mathcal{X}} \|2\Sigma_{\text{exact}}(x - x') - 2\Sigma_{\text{mis}}(x - x')\|_2 \tag{A.2}$$

$$\leq \sup_{x, x' \in \mathcal{X}} 2\|\Sigma_{\text{exact}} - \Sigma_{\text{mis}}\|_2 \cdot \|x - x'\|_2 \tag{A.3}$$

$$\leq D \cdot 2\|\Sigma_{\text{exact}} - \Sigma_{\text{mis}}\|_2. \tag{A.4}$$

The level of fair metric misspecification $\delta_d$ vanishes as long as $\Sigma_{\text{mis}}$ estimates $\Sigma_{\text{exact}}$ consistently.

**Theorem A.1** (Robustness of test statistic). *Suppose that the support of data distribution $P$ satisfies for any $(x_0, y_0) \in \text{supp}(P)$, the solution path of ODE (2.6) corresponding to $(x_0, y_0)$ and $d_1$ (or $d_2$) lies in $\mathcal{X}$. Under Assumptions A1 − A4, we have*

$$\left| \frac{\ell(f(\Phi_1(x_0, y_0)), y_0)}{\ell(f(x_0), y_0)} - \frac{\ell(f(\Phi_2(x_0, y_0)), y_0)}{\ell(f(x_0), y_0)} \right| \leq \sqrt{\frac{\lambda \delta_d}{L}} \frac{L_0 D e^{LT}}{c} \tag{A.5}$$

*for any $(x_0, y_0) \in \text{supp}(P)$.*

The first assumption in Theorem A.1 is mild since we always perform early termination in gradient flow attack.

In the literature, the fair metric can be learned from an additional dataset different from the training, test, and audit dataset. In this case, the constant $\delta_d$, which characterizes the goodness-of-fit of the estimated fair metric to the exact fair metric, shrinks to 0 as $n \to \infty$. Then Theorem A.1 provides two key insights.

First, as long as $\delta_d$ tends to 0, we ultimately test the same null hypothesis since

$$\left| \mathbb{E}_P \left[ \frac{\ell(f(\Phi_1(x_0, y_0)), y_0)}{\ell(f(x_0), y_0)} \right] - \mathbb{E}_P \left[ \frac{\ell(f(\Phi_2(x_0, y_0)), y_0)}{\ell(f(x_0), y_0)} \right] \right| \leq \sqrt{\frac{\lambda \delta_d}{L}} \frac{L_0 D e^{LT}}{c} \to 0 \tag{A.6}$$

as $\delta_d \to 0$.

Second, the error of test statistic induced by the misspecification of fair metric is *negligible* as long as $\delta_d = o(\frac{1}{n})$. This is due to the fact that the fluctuations of test statistic are $O_p(\frac{1}{\sqrt{n}})$, so $\sqrt{\delta_d}$ must vanish faster than $O(\frac{1}{\sqrt{n}})$ to not affect the test statistic asymptotically.

## B    PROOFS

### B.1    PROOF OF THEOREM IN SECTION 2

*Proof of Theorem 2.2.* Let $X(t) = (X^{(1)}(t), \ldots, X^{(d)}(t))^\top$. For $i = 1, \ldots, d$ and $k = 1, \ldots, N$, we have

$$X^{(i)}(t_k) = X^{(i)}(t_{k-1}) + \eta_k \dot{X}^{(i)}(t_{k-1}) + \frac{1}{2}\eta_k^2 \ddot{X}^{(i)}(\tilde{t}_{k-1}^{(i)}) \tag{B.1}$$

for some $\tilde{t}_{k-1}^{(i)} \in [t_{k-1}, t_k]$. Compactly, we have

$$X(t_k) = X(t_{k-1}) + \eta_k \dot{X}(t_{k-1}) + \frac{1}{2}\eta_k^2 \left( \ddot{X}^{(1)}(\tilde{t}_{k-1}^{(1)}), \dots, \ddot{X}^{(d)}(\tilde{t}_{k-1}^{(d)}) \right)^\top . \tag{B.2}$$

For $k = 1, \dots, N$, we let

$$T_k \triangleq \frac{X(t_k) - X(t_{k-1})}{\eta_k} - g(X(t_{k-1})) \tag{B.3}$$

$$= \frac{1}{\eta_k} \left( X(t_k) - X(t_{k-1}) - \eta_k \dot{X}(t_{k-1}) \right) . \tag{B.4}$$

Note that

$$\|\ddot{X}(t)\|_\infty = \|J_g(X(t))g(X(t))\|_\infty \le m, \tag{B.5}$$

and $\eta_k \le h$, we have

$$\|T_k\|_2 = \frac{1}{2}\eta_k \left\| \left( \ddot{X}^{(1)}(\tilde{t}_{k-1}^{(1)}), \dots, \ddot{X}^{(d)}(\tilde{t}_{k-1}^{(d)}) \right)^\top \right\|_2 \tag{B.6}$$

$$\le \frac{1}{2}\eta_k \sqrt{d} \left\| \left( \ddot{X}^{(1)}(\tilde{t}_{k-1}^{(1)}), \dots, \ddot{X}^{(d)}(\tilde{t}_{k-1}^{(d)}) \right)^\top \right\|_\infty \tag{B.7}$$

$$\le \frac{hm\sqrt{d}}{2} . \tag{B.8}$$

Let $e_k = X(t_k) - x^{(k)}$ for $k = 1, \dots, N$, we have

$$e_k = X(t_{k-1}) - x^{(k-1)} + \eta_k \left( g(X(t_{k-1})) - g(x^{(k-1)}) \right) + \eta_k T_k \tag{B.9}$$

$$= e_{k-1} + \eta_k \left( g(X(t_{k-1})) - g(x^{(k-1)}) \right) + \eta_k T_k . \tag{B.10}$$

Since $g$ is $L$-Lipschitz, we have

$$\|e_k\|_2 \le \|e_{k-1}\|_2 + \eta_k L \|e_{k-1}\|_2 + \eta_k \frac{hm\sqrt{d}}{2} . \tag{B.11}$$

Then,

$$\|e_k\|_2 + \frac{hm\sqrt{d}}{2L} \le (1 + L\eta_k) \left( \|e_{k-1}\|_2 + \frac{hm\sqrt{d}}{2L} \right) \tag{B.12}$$

$$\le e^{L\eta_k} \left( \|e_{k-1}\|_2 + \frac{hm\sqrt{d}}{2L} \right) . \tag{B.13}$$

For $k = 1, \dots, N$,

$$\|e_k\|_2 + \frac{hm\sqrt{d}}{2L} \le e^{L(\eta_1 + \cdots + \eta_k)} \frac{hm\sqrt{d}}{2L} \le e^{LT} \frac{hm\sqrt{d}}{2L} . \tag{B.14}$$

Therefore,

$$\max_{k=1,\dots,N} \|X(t_k) - x^{(k)}\|_2 = \max_{k=1,\dots,N} \|e_k\| \le \frac{hm\sqrt{d}}{2L}(e^{LT} - 1) . \tag{B.15}$$

$\square$

## B.2 Proof of Theorems and Corollaries in Section 3

*Proof of Theorem 3.1.* By central limit theorem (CLT),

$$\sqrt{n} \left( \mathrm{Var}_P \left[ \frac{\ell(f(\Phi(x,y)),y)}{\ell(f(x),y)} \right] \right)^{-\frac{1}{2}} \left( S_n - \mathbb{E}_P \left[ \frac{\ell(f(\Phi(x,y)),y)}{\ell(f(x),y)} \right] \right) \xrightarrow{d} \mathcal{N}(0,1) \tag{B.16}$$

Since

$$V_n^2 \xrightarrow{p} \mathrm{Var}_P \left[ \frac{\ell(f(\Phi(x,y)),y)}{\ell(f(x),y)} \right],$$ (B.17)

by Slutsky's theorem, we conclude that

$$\sqrt{n} V_n^{-1} \left( S_n - \mathbb{E}_P \left[ \frac{\ell(f(\Phi(x,y)),y)}{\ell(f(x),y)} \right] \right) \xrightarrow{d} \mathcal{N}(0,1).$$ (B.18)

$\square$

*Proof of Corollary 3.2.* By asymptotic distribution given by Theorem 3.1,

$$\mathbb{P} \left( \mathbb{E}_P \left[ \frac{\ell(f(\Phi(x,y)),y)}{\ell(f(x),y)} \right] \in \left[ S_n - \frac{z_{1-\alpha/2}}{\sqrt{n}} V_n, S_n + \frac{z_{1-\alpha/2}}{\sqrt{n}} V_n \right] \right)$$ (B.19)

$$= \mathbb{P} \left( z_{\alpha/2} \le \sqrt{n} V_n^{-1} \left( S_n - \mathbb{E}_P \left[ \frac{\ell(f(\Phi(x,y)),y)}{\ell(f(x),y)} \right] \right) \le z_{1-\alpha/2} \right) \to 1 - \alpha$$ (B.20)

as $n \to \infty$. $\square$

*Proof of Corollary 3.3.* Let $\tau = \mathbb{E}_P[\ell(f(\Phi(x,y)),y)/\ell(f(x),y)]$. By asymptotic distribution given by Theorem 3.1,

$$\mathbb{P}(T_n > \delta) = 1 - \mathbb{P} \left( S_n - \frac{z_{1-\alpha}}{\sqrt{n}} V_n \le \delta \right)$$ (B.21)

$$= 1 - \mathbb{P} \left( \sqrt{n} V_n^{-1} (S_n - \tau) \le z_{1-\alpha} + \sqrt{n} V_n^{-1} (\delta - \tau) \right)$$ (B.22)

$$\to \begin{cases} 0, & \text{if } \tau < \delta \\ \alpha, & \text{if } \tau = \delta \\ 1, & \text{if } \tau > \delta \end{cases}$$ (B.23)

as $n \to \infty$. $\square$

## B.3 PROOF OF THEOREM IN APPENDIX A

*Proof of Theorem A.1.* For any $(x_0, y_0) \in \mathcal{Z}$, let $\{X_1(t)\}_{0 \le t \le T}$ solve

$$\begin{cases} \dot{X}_1(t) = \nabla_x \{ \ell(f(X_1(t), y_0)) - \lambda d_1^2(X_1(t), x_0) \}, \\ X_1(0) = x_0, \end{cases}$$ (B.24)

and $\{X_2(t)\}_{0 \le t \le T}$ solve

$$\begin{cases} \dot{X}_2(t) = \nabla_x \{ \ell(f(X_2(t), y_0)) - \lambda d_2^2(X_1(t), x_0) \}, \\ X_2(0) = x_0. \end{cases}$$ (B.25)

Consider

$$y(t) = \|X_1(t) - X_2(t)\|_2^2 + \frac{\lambda D^2 \delta}{L},$$ (B.26)

we have

$$y(0) = \frac{\lambda D^2 \delta}{L}, \quad y(t) \ge 0,$$ (B.27)

and

$$\dot{y}(t) = 2 \langle X_1(t) - X_2(t), \dot{X}_1(t) - \dot{X}_2(t) \rangle$$ (B.28)

$$\le 2 \|X_1(t) - X_2(t)\|_2 \cdot \|\dot{X}_1(t) - \dot{X}_2(t)\|_2$$ (B.29)

$$\le 2 \|X_1(t) - X_2(t)\|_2 \cdot \{ L \|X_1(t) - X_2(t)\|_2 + \lambda D \delta \}$$ (B.30)

$$\le 2L \left\{ \|X_1(t) - X_2(t)\|_2^2 + \frac{\lambda D^2 \delta}{L} \right\}$$ (B.31)

$$= 2L \cdot y(t).$$ (B.32)

By Gronwall's inequality,

$$y(T) \le e^{2LT} y(0),$$ (B.33)

that is,

$$\|X_1(T) - X_2(T)\|_2^2 \le \frac{\lambda D^2 \delta}{L}(e^{2LT} - 1), \tag{B.34}$$

which implies

$$\|\Phi_1(x_0, y_0) - \Phi_2(x_0, y_0)\|_2 = \|X_1(T) - X_2(T)\|_2 \le \sqrt{\frac{\lambda \delta}{L}} D e^{LT}. \tag{B.35}$$

By Assumption A3, we have

$$\left| \frac{\ell(f(\Phi_1(x_0, y_0)), y_0)}{\ell(f(x_0), y_0)} - \frac{\ell(f(\Phi_2(x_0, y_0)), y_0)}{\ell(f(x_0), y_0)} \right| \le \sqrt{\frac{\lambda \delta}{L}} \frac{L_0 D e^{LT}}{c}. \tag{B.36}$$

$\square$

# C  ASYMPTOTIC NORMALITY AND ASYMPTOTIC VALIDITY OF THE ERROR-RATES RATIO TEST

The auditor collects a set of audit data $\{(x_i, y_i)\}_{i=1}^n$ and computes the ratio of empirical risks

$$\tilde{S}_n = \frac{\frac{1}{n}\sum_{i=1}^n \ell_{0,1}(f(\Phi(x_i, y_i)), y_i)}{\frac{1}{n}\sum_{i=1}^n \ell_{0,1}(f(x_i), y_i)} \text{ and } \tilde{V}_n \triangleq \frac{1}{n}\sum_{i=1}^n \begin{bmatrix} \ell_{0,1}(f(\Phi(x_i, y_i)), y_i) \\ \ell_{0,1}(f(x_i), y_i) \end{bmatrix} \begin{bmatrix} \ell_{0,1}(f(\Phi(x_i, y_i)), y_i) \\ \ell_{0,1}(f(x_i), y_i) \end{bmatrix}^\top \tag{C.1}$$

by performing the gradient flow attack (2.6). Let $A_n$ and $B_n$ be the numerator and denominator of $\tilde{S}_n$.

First we derive the limiting distribution of a calibrated test statistic for the error-rates ratio test.

**Theorem C.1** (Asymptotic distribution). *Assume that $\nabla_x\{\ell(f(x), y) - \lambda d_x^2(x, y)\}$ is Lipschitz in $x$. If the ML model $f$ is independent of $\{(x_i, y_i)\}_{i=1}^n$, then*

$$\frac{\sqrt{n}B_n^2}{\sqrt{A_n^2(\tilde{V}_n)_{2,2} + B_n^2(\tilde{V}_n)_{1,1} - 2A_nB_n(\tilde{V}_n)_{1,2}}} \left( \tilde{S}_n - \frac{\mathbb{E}_P[\ell_{0,1}(f(\Phi(x, y)), y)]}{\mathbb{E}_P[\ell_{0,1}(f(x), y)]} \right) \xrightarrow{d} \mathcal{N}(0, 1) \tag{C.2}$$

*in distribution, as $n \to \infty$.*

Type I error rate control is formalized in the following:

**Corollary C.2** (Asymptotic validity of test). *Under the assumptions in Theorem C.1,*

1. *if $\mathbb{E}_P[\ell_{0,1}(f(\Phi(x, y)), y)]/\mathbb{E}_P[\ell_{0,1}(f(x), y)] \le \delta$, we have $\lim_{n\to\infty} \mathbb{P}(\tilde{T}_n > \delta) \le \alpha$;*
2. *if $\mathbb{E}_P[\ell_{0,1}(f(\Phi(x, y)), y)]/\mathbb{E}_P[\ell_{0,1}(f(x), y)] > \delta$, we have $\lim_{n\to\infty} \mathbb{P}(\tilde{T}_n > \delta) = 1$.*

*Proof of Theorem C.1.* By central limit theorem (CLT),

$$\sqrt{n}\left( \begin{bmatrix} A_n \\ B_n \end{bmatrix} - \begin{bmatrix} \mu_x \\ \mu_y \end{bmatrix} \right) \xrightarrow{d} \mathcal{N}(0, \Sigma), \tag{C.3}$$

for finite $\mu_x, \mu_y$, and $\Sigma$ with finite entries. Let $g(x, y) = x/y$, then $\nabla g(\mu_x, \mu_y) = \mu_y^{-2}(\mu_y, -\mu_x)^\top$. By continuous mapping theorem, we have

$$\sqrt{n}\left(g(A_n, B_n) - g(\mu_x, \mu_y)\right) \xrightarrow{d} \mathcal{N}(0, \nabla g(\mu_x, \mu_y)^\top \Sigma \nabla g(\mu_x, \mu_y)), \tag{C.4}$$

which implies

$$\sqrt{n}\left( \tilde{S}_n - \frac{\mathbb{E}_P[\ell_{0,1}(f(\Phi(x, y)), y)]}{\mathbb{E}_P[\ell_{0,1}(f(x), y)]} \right) \xrightarrow{d} \mathcal{N}\left( 0, \frac{\mu_x^2 \Sigma_{2,2} + \mu_y^2 \Sigma_{1,1} - 2\mu_x\mu_y\Sigma_{1,2}}{\mu_y^4} \right), \tag{C.5}$$

or

$$\frac{\sqrt{n}\mu_y^2}{\sqrt{\mu_x^2\Sigma_{2,2} + \mu_y^2\Sigma_{1,1} - 2\mu_x\mu_y\Sigma_{1,2}}} \left( \tilde{S}_n - \frac{\mathbb{E}_P[\ell_{0,1}(f(\Phi(x, y)), y)]}{\mathbb{E}_P[\ell_{0,1}(f(x), y)]} \right) \xrightarrow{d} \mathcal{N}(0, 1). \tag{C.6}$$

Since $A_n \overset{p}{\to} \mu_x$, $B_n \overset{p}{\to} \mu_y$ and $\tilde{V}_n \overset{p}{\to} \Sigma$, we therefore conclude by Slutsky's Theorem that

$$\frac{\sqrt{n}B_n^2}{\sqrt{A_n^2(\tilde{V}_n)_{2,2} + B_n^2(\tilde{V}_n)_{1,1} - 2A_nB_n(\tilde{V}_n)_{1,2}}} \left(\tilde{S}_n - \frac{\mathbb{E}_P[\ell_{0,1}(f(\Phi(x,y)),y)]}{\mathbb{E}_P[\ell_{0,1}(f(x),y)]}\right) \overset{d}{\to} \mathcal{N}(0,1).$$

$\square$

*Proof of Corollary C.2.* Let $\tau = \mathbb{E}_P[\ell_{0,1}(f(\Phi(x,y)),y)]/\mathbb{E}_P[\ell_{0,1}(f(x),y)]$. By asymptotic distribution given by Theorem C.1,

$$\mathbb{P}(\tilde{T}_n > \delta) = 1 - \mathbb{P}\left(\tilde{S}_n - \frac{z_{1-\alpha}}{B_n^2}\sqrt{\frac{A_n^2(\tilde{V}_n)_{2,2} + B_n^2(\tilde{V}_n)_{1,1} - 2A_nB_n(\tilde{V}_n)_{1,2}}{n}} \le \delta\right)$$

$$= 1 - \mathbb{P}\left(\frac{\sqrt{n}B_n^2}{\sqrt{A_n^2(\tilde{V}_n)_{2,2} + B_n^2(\tilde{V}_n)_{1,1} - 2A_nB_n(\tilde{V}_n)_{1,2}}}(\tilde{S}_n - \tau)\right.$$

$$\left.\le z_{1-\alpha} + \frac{\sqrt{n}B_n^2}{\sqrt{A_n^2(\tilde{V}_n)_{2,2} + B_n^2(\tilde{V}_n)_{1,1} - 2A_nB_n(\tilde{V}_n)_{1,2}}}(\delta - \tau)\right)$$

$$\to \begin{cases} 0, & \text{if } \tau < \delta \\ \alpha, & \text{if } \tau = \delta \\ 1, & \text{if } \tau > \delta \end{cases}$$

as $n \to \infty$. $\square$

## D    IMPLEMENTATION OF THE PROPOSED TEST

The algorithm 1 provides a step-by-step procedure for calculating the lower bound. For a choice of $\delta$ (threshold for null hypothesis, see equation 3.3), at a level of significance 0.05, we reject the null hypothesis if lower bound is greater than $\delta$.

---

**Algorithm 1** Individual fairness testing

---

**Input**: ML model $f$; loss $\ell$; data $\{(X_i, Y_i)\}_{i=1}^n$; fair-distance $d_x$; regularization parameters $\lambda$; number of steps $T$; and step size $\{\epsilon_t\}_{t=1}^T$;
**Require**: $f$ provides class probabilities; $\epsilon_t$ is decreasing
**for** $i = 1, \ldots, n$ **do**
    Initialize $X_i' \leftarrow X_i$
    **for** $t = 1, \ldots, T$ **do**
        $X_i' \leftarrow X_i' + \epsilon_t \nabla \{\ell(f(X_i'), Y_i) - \lambda d_x(X_i', X_i)\}$
    **end for**
    $r_i \leftarrow \frac{\ell(f(X_i'), Y_i)}{\ell(f(X_i), Y_i)}$
**end for**

**Output**: lower bound $= \text{mean}(r) - \frac{1.645}{\sqrt{n}} * \text{std}(r)$

---

## E    SUPPLEMENTARY DETAILS FOR SIMULATIONS

Here we provide further details for the experiment with simulated data.

### E.1    DATA

We considered one one group variable $G$ with two labels. The 2 dimensional features were generated with the idea that they will differ in first co-ordinate. We present the detailed model for generating

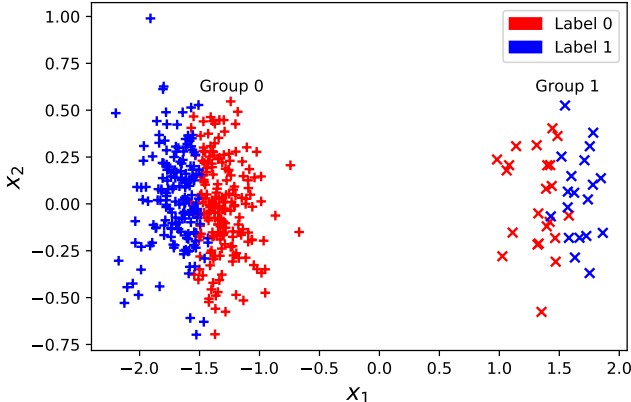

Figure 3: Simulated data for synthetic experiment. We have two groups; group 1 being the minority group. The features for these two groups differ mainly by first co-ordinate. So, the discounted movement direction is $(1,0)^T$.

the data.

$$
\begin{cases}
G_i \sim \text{iid Bernoulli}(0.1) \\
X_i \sim \mathcal{N}\left((1-G_i)\begin{bmatrix} -1.5 \\ 0 \end{bmatrix} + G_i \begin{bmatrix} 1.5 \\ 0 \end{bmatrix}, (0.25)^2 \mathbf{I}_2\right) \\
Y_i = \mathbb{1}\left\{\left((1-G_i)\begin{bmatrix} -0.2 \\ -0.01 \end{bmatrix} + G_i \begin{bmatrix} 0.2 \\ -0.01 \end{bmatrix}\right)^T X_i + \mathcal{N}\left(0, 10^{-4}\right) > 0\right\} \\
\qquad \text{for } i = 1, \dots, 400.
\end{cases}
\tag{E.1}
$$

The data is plotted in Figure 3. As seen in the figure, feature vectors for two groups mainly differ in the first co-ordinate. So, the discounted movement direction is $(1,0)^T$.

## E.2 CLASSIFIERS

For comparison purpose we consider logistic model of the form

$$
f_{b,w_1,w_2}(x) = \text{expit}\left(b + w_1 X_i^{(1)} + w_2 X_i^{(2)}\right)
\tag{E.2}
$$

where $\text{expit}(x) \triangleq \frac{e^x}{1+e^x}$ and the weights are chosen as $(w_1, w_2) \in \{-4, -3.6, \dots, 4\}^2$. For a given $(w_1, w_2)$ the bias $b$ is chosen as:

$$
b(w_1, w_2) \triangleq \arg\min_{b \in \mathbf{R}} \sum_{i=1}^{400} \ell(f_{b,w_1,w_2}(X_i), Y_i),
$$

where $\ell$ is the logistic loss.

## E.3 LOWER BOUND

To calculate the lower bounds we use Algorithm 1 with the following choices: the choices of $\ell$ and $f$ is provided in the previous subsection. The choice of fair distances are provided in Section 4. We choose regularizer $\lambda = 100$, number of steps $T = 400$ and step sizes $\epsilon_t = \frac{0.02}{t^{2/3}}$.

Table 2: Choice of hyperparameters for Baseline and Project

| Parameters | learning_rate | batch_size | num_steps |
|:---:|:---:|:---:|:---:|
| Choice | $10^{-4}$ | 250 | 8K |

## F  ADDITIONAL ADULT EXPERIMENT DETAILS

### F.1  DATA PREPROCESSING

The continuous features in `Adult` are: `Age, fnlwgt, capital-gain, capital-loss, hours-per-week`, and `education-num`. The categorical features are: `work-class, education, marital-status, occupation, relationship, race, sex, native-country`. The detailed descriptions can be found in Dua & Graff (2017). We remove `fnlwgt, education, native-country` from the features. `race` and `sex` are considered as protected attributes and they are not included in feature vectors for classification. `race` is treated as binary attribute: White and non-White. We remove any data-point with missing entry and end up having 45222 data-points.

### F.2  FAIR METRIC

To learn the sensitive subspace, we perform logistic regression for `race` and `sex` on other features, and use the weight vectors as the vectors spanning the sensitive subspace ($\mathcal{H}$). The fair metric is then obtained as

$$d_x^2(x_1, x_2) = \|(I - \Pi_{\mathcal{H}})x_1 - x_2\|_2^2.$$

### F.3  HYPERPARAMETERS AND TRAINING

For each model, 10 random train/test splits of the dataset is used, where we use $80\%$ data for training purpose. All compared methods are adjusted to account for class imbalance during training.

#### F.3.1  BASELINE AND PROJECT

Baseline is the obtained by fitting 2 layer fully connected neural network with 50 neurons for the hidden layer. It doesn't enforce any fairness for the model. Project also uses similar architecture, except a pre-processing layer for projecting out sensitive subspace from features. So, Project model is a simple and naive way to enforce fairness. For both the models same parameters are involved: `learning_rate` for step size for `Adam` optimizer, `batch_size` for mini-batch size at training time, and `num_steps` for number of training steps to be performed. We present the choice of hyperparameters in Table 2

#### F.3.2  SENSR

Codes for SenSR (Yurochkin et al., 2020) is provided with submission with a demonstration for fitting the model, where the choice of hyperparameters are provided.

#### F.3.3  REDUCTION

We provide codes for reduction (Agarwal et al., 2018) approach. We also provide a demonstration for fitting reduction model with the choice of hyperparameters for this experiment. The codes can also be found in `https://github.com/fairlearn/fairlearn`. We used Equalized Odds fairness constraint (Hardt et al., 2016) with constraints violation tolerance parameter $\epsilon = 0.03$.

### F.4  LOWER BOUND AND TESTING

To calculate the lower bounds we use Algorithm 1. The loss $\ell$ is the logistic loss. Test data is provided as an input, whereas the fair metric is also learnt from the test data. For each of the models we choose regularizer $\lambda = 50$, number of steps $T = 500$ and step size $\epsilon_t = 0.01$.

Table 3: Full results on `ADULT` data over 10 experiment repetitions

|  |  | Baseline | Project | Reduction | SenSR |
|---|---|---|---|---|---|
|  | bal-acc | 0.817±0.007 | **0.825**±0.003 | 0.800±0.005 | 0.765±0.012 |
|  | $\text{AOD}_{\text{gen}}$ | -0.151±0.026 | -0.147±0.015 | **0.001**±0.021 | -0.074±0.033 |
|  | $\text{AOD}_{\text{race}}$ | -0.061±0.015 | -0.053±0.015 | **-0.027**±0.013 | -0.048±0.008 |
| Entropy loss | $T_n$ | 3.676±2.164 | 1.660±0.355 | 5.712±2.264 | **1.021**±0.008 |
|  | reject-prop | 1.0 | 0.9 | 1.0 | **0.0** |
| 0-1 loss | $\tilde{T}_n$ | 2.262±0.356 | 1.800±0.584 | 3.275±0.343 | **1.081**±0.041 |
|  | reject-prop | 1.0 | 0.8 | 1.0 | **0.0** |

### F.5 COMPARISON METRICS

**Performance** Let $\mathcal{C}$ be the set of classes. Let $Y$ and $\hat{Y}$ be the observed and predicted label for a data-point, respectively. The balanced accuracy is defined as

$$\text{Balanced Acc} = \frac{1}{|\mathcal{C}|} \sum_{c \in \mathcal{C}} P(\hat{Y} = c | Y = c)$$

**Group fairness** Let $G$ be the protected attribute taking values in $\{0, 1\}$. The average odds difference (AOD) (Bellamy et al., 2018) for group $G$ is defined as

$$\text{AOD}_G = \frac{1}{2} \Big[ (P(\hat{Y} = 1 | Y = 1, G = 1) - P(\hat{Y} = 1 | Y = 1, G = 0))$$
$$+ (P(\hat{Y} = 1 | Y = 0, G = 1) - P(\hat{Y} = 1 | Y = 0, G = 0)) \Big]$$

### F.6 FULL TABLE

In Table 3 we present extended results of the `Adult` experiment with standard deviations computed from 10 experiment repetitions.

## G COMPAS EXPERIMENT

In `COMPAS` recidivism prediction dataset (Larson et al., 2016) the task is to predict whether a criminal defendant would recidivate within two years. We consider `race` (Caucasian or not-Caucasian) and `sex` (binary) as the sensitive attributes. The features in `COMPAS` are: `sex, race, priors_count age_cat= 25 to 45, age_cat= Greater than 45, age_cat= Less than 25,` and `c_charge_degree=F`. `prior_count` is standardized.

As before we perform testing on four classifiers: baseline NN, group fairness Reductions (Agarwal et al., 2018) algorithm, individual fairness SenSR (Yurochkin et al., 2020) algorithm and a basic Project algorithm that pre-processes the data by projecting out the "senstive" subspace. Baseline and Project have same architecture and parameters as in the experiment with `Adult` dataset. For SenSR fair metric and Project we use train data to learn the "senstive" subspace. A further detail for choice of parameters is provided in the code. For Reduction we used Equalized Odds fairness constraint (Hardt et al., 2016) with constraints violation tolerance parameter $\epsilon = 0.16$.

All methods are trained to account for class imbalance in the data and we report test balanced accuracy as a performance measure. Results of the 10 experiment repetitions are summarized in Table 4. We compare group fairness using average odds difference (AOD) (Bellamy et al., 2018) for gender and race. Significance level for the null hypothesis rejection is 0.05 and $\delta = 1.25$.

Baseline exhibits clear violations of both individual (test is rejected with proportion 1) and group fairness (both AODs are big in terms of absolute magnitude). Reductions method achieves significant group fairness improvements, but is individually unfair. Simple pre-processing is more efficient (comparing to the `Adult` experiment) with rejection proportion of 0.2. SenSR is the most effective

Table 4: Results on COMPAS data over 10 experiment repetitions

|  | Balanced Acc | $AOD_{gen}$ | $AOD_{race}$ | $T_n$ | Rejection Prop |
|---|---|---|---|---|---|
| Baseline | **0.675**±0.013 | 0.218±0.041 | 0.260±0.026 | 2.385±0.262 | 1.0 |
| Project | 0.641±0.017 | 0.039±0.029 | 0.227±0.021 | 1.161±0.145 | 0.2 |
| Reduction | 0.652±0.012 | **-0.014**±0.054 | **0.037**±0.039 | 1.763±0.069 | 1.0 |
| SenSR | 0.640±0.022 | 0.046±0.031 | 0.237±0.018 | **1.098**±0.061 | **0.0** |

and our test fails to reject its individual fairness in all experiment repetitions. Examining the trade-off between individual and group fairness, we see that both SenSR and Reductions improve all fairness metrics in comparison to the baseline. However, improvement of individual fairness with Reductions is marginal. SenSR provides a sizeable improvement of gender AOD, but only a marginal improvement of race AOD.

