# OpenReview forum: "Statistical inference for individual fairness"
_ICLR.cc/2021/Conference — ICLR 2021 Poster_

### Official Review · AnonReviewer2 · 2020-10-27

**Rating:** 6
**Confidence:** 3

**Review:**

This paper proposes a test to determine whether an ML model violates individual fairness. The main contribution beyond existing work is that this method allows for continuous feature spaces.

Conceptually, this paper rests on the "gradient flow attack," which produces a mapping that, given an example, produces another example which violates the inviddual fairness constraint. Thus, given a distribution, one can compute the change in loss between the original distribution and the mapped distribution. The ratio of these quantities is what the authors use for their hypothesis testing problem: is it below a the limit of tolerance or not?

For the most part, the paper is farily well-written, though it gets a little more difficult to understand towards the end. The basic premise is interesting -- using a gradient flow attack to discover pairs of elements that are similar but have different predicted outcomes.

I don't fully understand the motivation behind the loss ratio statistic. Why compare $\ell(f(\Phi(x, y)), y)$ and $\ell(f(x), y)$ instead of simply $f(\Phi(x, y))$ and $f(x)$? Does this become very sensitive when $\ell(f(x), y)$ is close to 0? If so, then it seems as though worse models might more easily pass the test, since the denominator of the loss ratio would generally be larger. I think the gradient flow attack is promising here, but I'm not convinced that this is the right test to run.

I'm not completely convinced by the claim that this test is interpretable: while the authors point to the 4/5 rule as a measure of impact, the loss ratio proposed here is clearly measuring something quite different from what a disparate impact test would traditionally measure. I don't see this as making the test results interpretable.

---

> ### Author Response · Authors · 2020-11-23
> **Response to Reviewer 2**
>
> We thank you for the feedback. Please see our general response for the summary of key changes. We address your questions and concerns below.
>
> **Why compare $\ell(f(\Phi(x,y)),y)$ and $\ell(f(x),y)$ instead of $f(\Phi(x,y))$ and $f(x)$?**
>
> We compare the losses because we are auditing the model for performance differentials across similar samples, and we are using the loss function as a measure of performance. It is certainly possible to compare $f(\Phi(x,y))$ and $f(x)$ as long as the user defines a suitable metric on the output space of the ML model.
>
> **Does this become very sensitive when $\ell(f(x),y)$ is close to zero.**
>
> If the loss is small but non-zero, then the variance of the test statistic is inflated and the test loses power but maintains Type I error rate control. We added a comment to section 3 describing the drawbacks of using the ratio of losses as a test statistic. Please also see Section 3.3 for an alternative test statistic based on the error-rate ratio.
>
> **I'm not completely convinced by the claim that this test is interpretable: while the authors point to the 4/5 rule as a measure of impact, the loss ratio proposed here is clearly measuring something quite different from what a disparate impact test would traditionally measure. I don't see this as making the test results interpretable.**
>
> The claim of interpretability applies to the choice of the threshold parameter $\delta$ and its interpretation as a ratio of model performance on counterfactual data (generated with the gradient flow attack) and the original data. We brought up the 4/5 rule as an example of one way to think of this parameter. For specific ML tasks, there may be more appropriate settings of this tolerance parameter. For further discussion regarding interpretability and our usage of the 4/5 rule, please see an extended discussion in Section 3.3 and an alternative test statistic that we propose there. It is based on the error-rates ratio and has closer connection to the 4/5 rule. Both loss ratio and error-rates ratio tests perform similarly in practice as we show in the updated Table 1.
>
> More broadly, we wish to point out that any auditing/testing procedure requires user to pick some sort of threshold. Appropriate choice of this threshold is highly application specific and it is impossible to provide a general scientifically-justified threshold. For the purposes of the paper, we could have picked an arbitrary $\delta$ value but instead we decided to draw inspiration from an existing regulation. Four-fifth rule applies to group fairness (i.e. disparate impact, as you noted), but unfortunately this is the only existing legal regulation relevant to algorithmic fairness. Given the importance of algorithmic fairness, we hope that in the future policy-makers will consider individual fairness and formulate appropriate regulations. Our testing methodology is quite flexible and is likely to be possible to adapt to such regulations.

---

### Official Review · AnonReviewer4 · 2020-10-29
**Weak Accept**

**Rating:** 6
**Confidence:** 4

**Review:**

The paper introduces a framework to statistically test whether a given model is individually fair or not. In particular, given a model, a distance metric over individuals, and a data point z, the authors propose an algorithm that finds a new data point z' such that z' is similar to z but their corresponding losses are different under the model -- if the model is not individually fair. They provide experimental results to show how their proposed method can detect unfairness in practice.

I think the paper tackles an interesting problem and has nice results, both theoretically and experimentally. My major concern about this paper is a fundamental one: what do you exactly mean by "individual fairness"? I think there should be a formal definition for the fairness notion you have in mind. According to the Individual Fairness notion of Dwork et al., even one couple of similar examples on which the given model performs differently constitutes unfairness. But it looks like the fairness notion in this paper requires that *on average over the input data distribution* the model is treating similar individuals similarly which is different/weaker than the notion proposed by Dwork et al.. Please clarify if I'm missing something, or else formally define the fairness notion you used in this work.

Other comments which are mostly about the technical development early on in the paper that I find hard to follow:

-I don't quite understand Eq. 2.2 and how it is solving an individually fair learning problem. Isn't that just the maximum expected loss on distributions that are epsilon far from the empirical distribution? If so, how does this help with fairness? Should W(P,P_n) be defined somewhere?

-Also, how is the dual problem obtained in Eq. 2.3? The authors say “it is known” but I think this requires more explanation/derivation.

-In Eq. 2.4, the function \ell_\lambda^c is defined as getting f(x_i) (a label) as input, but looking at the right hand side of the equation, this function actually depends on x_i itself, not f(x_i).

-How is the gradient flow attack related to the dual problem in Eq. 2.3 and 2.4? What happened to “epsilon” in this continuous formulation? It looks like that the primary objective now is to solve Eq. 2.4, and not the actual dual problem in 2.3, right? If that’s the case then what happened to primal and dual problems?

-How should one pick the stopping time T in Eq. 2.6 and how that affects the proposed method for finding the unfair map? Shouldn’t there be a theoretical statement about X(T), or the unfair map?

Overall I found section 2 of the paper very confusing. I will raise my score if the issues raised above are addressed.

---

> ### Author Response · Authors · 2020-11-23
> **Response to Reviewer 4**
>
> We thank the reviewer for the questions and recommendations for improvement. Please see our general response for the summary of the key changes. In particular, we have added a discussion and empirical study of the effect of stopping time as you suggested. Please see our responses below.
>
> **My major concern about this paper is a fundamental one: what do you exactly mean by "individual fairness"? I think there should be a formal definition for the fairness notion you have in mind.**
>
> We are using a risk-based notion of individual fairness that requires an ML model to have similar accuracy/performance on similar samples. This is a variant of Dwork et al's original definition of individual fairness that was first proposed in Yurochkin et al (2020). We added a formal definition of this variant of individual fairness in section 2 (Definition 2.1).
>
> **How should one pick the stopping time T in Eq. 2.6 and how that affects the proposed method for finding the unfair map? Shouldn’t there be a theoretical statement about X(T), or the unfair map?**
>
> We have added a discussion and an empirical study of the stopping time $T$ in section 4.2. Theoretically, Corollary 3.1 guarantees Type I error control for any $T$. It is hard to control Type II error in general - one needs to know expected value of the loss ratio as a function of $T$, which is model specific. In Figure 2 we show empirically that picking a sufficiently large $T$ helps to reduce Type II error.
>
> **Questions about technical development:**
>
> - Eq 2.2 appears in the definition of distributionally robust fairness (DRF). This is a variant of individual fairness that requires the accuracy/performance of a model to be similar on similar inputs. $W$ is the Wasserstein distance on probability distributions on feature space induced by the fair metric, and $P_n$ is the empirical distribution of the training data, so $W(P,P_n)$ is the Wasserstein distance between $P$ and $P_n$ ($P$ is the optimization variable in Eq 2.2).
> - We added a citation for the derivation of the dual problem in Eq 2.3.
> - Thanks for pointing out this notational inconsistency. We have fixed it in a revised version.
> - The gradient flow in Eq 2.6 is the gradient flow for evaluating the maximum in Eq 2.5. We are using the dual problem (Eq 2.4) to motivate the definition of the gradient flow. Once the gradient flow is defined, we are ignoring the dual problem (including its parameters $\epsilon$) and solely working with the gradient flow in the subsequent developments.

---

### Official Review · AnonReviewer1 · 2020-10-29
**Interesting direction, but design choices can use more work**

**Rating:** 6
**Confidence:** 3

**Review:**

The paper focuses on detecting "individual unfairness" in supervised learning.

The main contributions are:
1.  A method to generate "adversarial" examples, that is, the examples that are very close to the original input, but get a very different outcome. An optimization problem is used formulated by leveraging the DRO framework. The authors then point out the difficulty in solving the problem (specially with continuous features) and propose an ODE based solution to find the adversarial examples.
2. After finding the adversarial examples, a hypothesis testing framework is proposed to test the model for individual unfairness. The main idea here to compute the mean and variance of the test statistic on a given set of data points and then construct the confidence intervals using the Normality assumption.

On a high level, the idea of testing for individual unfairness is an interesting one, specially given that there aren't many metrics of it individual unfairness out there. However, it feels like many important design choices are not very clear. For these reasons, this reviewer is split between a weak accept and a weak reject. See the detailed comments below:

1. Intuitively, it seems like the need for hypothesis testing arises when one is working with a small test set (if the test set is large enough, then assuming IID samples, one could already be quite confident of the point estimate of the amount of individual unfairness as measured in Eq. 3.2). However, the paper then assumes that the distance metric is learnt from the test set. Now if the test set is already quite small, how good a metric do we expect to learn? It is not clear how to reconcile these two problems.

2. How much "interpretability" does the hypothesis test really add? The test statistic does really provide a whole lot more interpretability on on top of the quantity measured in Eq. 3.2 (which itself it very closely related to that in Eq. 2.4). Essentially, the main insight seems to be to monitor the change in loss from an example to an adversarial version of it. So that additional benefit does the hypothesis testing bring here, specially when working with reasonably sized test sets (also, see the point about the need for hypothesis testing above)? Given that the paper does not offer any discussion into the tradeoff between type I and II errors, it is not clear that advantages does the hypothesis testing bring for us.

3. Perhaps it would be worth discussing how the scale of the loss (<<0) might make the ratio in Eq. 3.2 unstable in practice.

4. This reviewer is a bit confused about the Normality result derived in Theorem 3.1. True that the distribution here tends to a Normal when $n \to \infty$. However, with small test sets, how well does this assumption hold? If it does not, the interval constructed in Eq. 3.6 may not be very accurate. In general, an empirical analysis of the intervals as in (https://arxiv.org/pdf/2007.05124.pdf) would be a great addition to the paper.

5. The paper seems to take the assumption that the distance metric specified by the user is differentiable (for solving the problem in Section 2). Is that true? It is quite possible for domain experts to specify distance metrics with discontinuities in them (e.g., if education level of x1 is higher than education level of x2, upweigh the distance by 1). Can such user-specified metrics be handled by the methods in the paper?

6. This reviewer is not very sure about the usage of four-fifth rule for the hypothesis test. While I am not a legal expert, the four-fifth rule seems to apply to groups instead of individuals as suggested by the paper. Moreover, applying the four-fifth rule on well-understood and well-bounded quantities (acceptance rate of the two groups), as is done in the group fairness literature, indeed makes sense. However, applying the same ratio threshold on a quantity such as loss that can be arbitrarily high or low might not be very interpretable (also see point 3). For instance, if the original loss is very low (<<0) or very high (in the order of 10's), does it make sense to apply the ratio test. Similarly, the test might lead to very different behavior when the loss changes from say hinge loss to squared loss to logistic loss. Is this behavior indeed desirable? Some explanation here would be greatly helpful in convincing the readers of the usefulness of the ratio test.

------------------

Post-rebuttal comments: Thanks for the detailed answers. Many of my concerns were addressed, and I am increasing my score as a result. A follow-up thought: It would be nice to add some discussion on the runtime of the proposed framework.

---

> ### Author Response · Authors · 2020-11-23
> **Response to Reviewer 1 [Part 1]**
>
> We thank the reviewer for the feedback. Please see our general response for the summary of key changes. We address questions and concerns below.
>
> **However, the paper then assumes that the distance metric is learnt from the test set. Now if the test set is already quite small, how good a metric do we expect to learn? It is not clear how to reconcile these two problems.**
>
> We study the sensitivity of the test statistic to estimation errors in the fair metric in section 3.2: the bottom line is the test statistic is not very sensitive to such errors. We also would like to clarify that we do not assume that the fair metric is learned from the test/audit set. Fair metric learning is a separate problem studied in the literature and there are multiple ways to perform this task using various forms of supervision. In other words, testing and fair metric learning are independent problems.
>
> **Essentially, the main insight seems to be to monitor the change in loss from an example to an adversarial version of it. So that additional benefit does the hypothesis testing bring here, specially when working with reasonably sized test sets (also, see the point about the need for hypothesis testing above)?**
>
> Adversarial attacks always increase the loss, even on a perfectly fair ML model. The main benefit of our approach is a statistically principled way of setting a threshold for deciding whether the increase is large enough to indicate algorithmic bias. In our approach, this threshold is set in a way that ensures adversarial attacks on fair ML models do not increase the loss beyond this threshold $(1-\alpha)$-fraction of the time (the significance level $\alpha$ can be set by the user).
>
> **Given that the paper does not offer any discussion into the tradeoff between type I and II errors, it is not clear that advantages does the hypothesis testing bring for us.**
>
> In this paper, we adopt the (standard) Neyman-Pearson approach to statistical hypothesis testing: we maximize power (minimize Type II error rate) subject to a constraint on the Type I error rate. That said, the benefit of hypothesis testing is it offers a statistically principled way to decide whether the increase in loss from the adversarial attack is large enough to indicate algorithmic bias.
>
> **Perhaps it would be worth discussing how the scale of the loss (<<0) might make the ratio in Eq. 3.2 unstable in practice.**
>
> We added a comment to section 3 describing the drawbacks of using the ratio of losses as a test statistic. If the loss is small but non-zero, then the variance of the test statistic is inflated and and the test loses power, but Type I error rate control is maintained. Please also see an alternative test statistic based on the error-rates ratio that we propose in Section 3.3.
>
> **However, with small test sets, how well does this [asymptotic normality] assumption hold? If it does not, the interval constructed in Eq. 3.6 may not be very accurate.**
>
> The asymptotic result that we are appealing to here is a central limit theorem. If the test set consists of independent test examples, then we expect the normal approximation to be accurate if there are at least 20 to 30 test examples.
>
> **The paper seems to take the assumption that the distance metric specified by the user is differentiable. Is that true? It is quite possible for domain experts to specify distance metrics with discontinuities in them?**
>
> The reviewer is correct in pointing out that we assume the fair metric is smooth. It is required for the gradient flow attack to be well-defined. In practice, if domain expert provides a discontinuous metric, it can be distilled into a smooth one.

---

> > ### Author Response · Authors · 2020-11-23
> > **Response to Reviewer 1 [Part 2]**
> >
> > **Moreover, applying the four-fifth rule on well-understood and well-bounded quantities (acceptance rate of the two groups), as is done in the group fairness literature, indeed makes sense. However, applying the same ratio threshold on a quantity such as loss that can be arbitrarily high or low might not be very interpretable.**
> >
> > We thank the reviewer for this comment. The new test statistic utilizing error-rates ratio that we propose in Section 3.3 addresses this concern. In practice, both loss ratio and error-rates ratio tests behave almost identically as can be seen in the updated Table 1.
> >
> > **This reviewer is not very sure about the usage of four-fifth rule for the hypothesis test. While I am not a legal expert, the four-fifth rule seems to apply to groups instead of individuals as suggested by the paper.**
> >
> > Four-fifth rule indeed applies to group fairness, but unfortunately this is the only existing legal regulation relevant to algorithmic fairness. Any auditing/testing procedure requires user to pick some sort of threshold. Appropriate choice of this threshold is highly application specific and it is impossible to provide a general scientifically-justified threshold. For the purposes of the paper, we could have picked an arbitrary $\delta$ value but instead we decided to draw inspiration from the only existing regulation. Given the importance of algorithmic fairness, we hope that in the future policy-makers will consider individual fairness and formulate appropriate regulations. Our testing methodology is quite flexible and is likely to be possible to adapt to such regulations.

---

### Author Response · Authors · 2020-11-23
**Summary of the key changes**

We thank the reviewers for their feedback.

**Below we provide the summary of the main changes.**

* We added Section 3.3 where we extend the discussion regarding interpretability of the test and propose a variant of the test statistic based on the error-rates ratio specific to classification. Both tests (using loss ratios and error-rates ratio) perform similarly in practice as shown in the updated Table 1. These changes were motivated by the comments of Reviewer 1 and Reviewer 2 regarding the interpretability of the loss ratio in the analogy to the four-fifths rule we used for defining the test threshold $\delta$.

* We added Figure 2 empirically studying the effect of the number of steps $T$ as suggested by Reviewer 4. Our Corollary 3.3 guarantees Type I error control for any $T$, however $T$ can effect the Type II error. Our empirical study provides insights into this connection.

---

### Decision · Program_Chairs · 2021-01-07
**Final Decision**

**Decision:**

Accept (Poster)

**Comment:**

This paper studies how to statistically test if a given model violates the constraint of individual fairness. This is an interesting and novel problem, and the paper leverages the technique of gradient flow to identify a "witness" pair for individual fairness violation.
During the rebuttal, the authors have addressed many concerns raised in the reviews. The author should also consider discussing the runtime and improving the exposition to resolve some of the presentation issues raised in the reviews.